# Determinants of Food Choice and Perceptions of Supermarket-Based Nudging Interventions among Adults with Low Socioeconomic Position: The SUPREME NUDGE Project

**DOI:** 10.3390/ijerph18116175

**Published:** 2021-06-07

**Authors:** Marjolein C. Harbers, Cédric N.H. Middel, Josine M. Stuber, Joline W.J. Beulens, Femke Rutters, Yvonne T. van der Schouw

**Affiliations:** 1Julius Center for Health Sciences and Primary Care, University Medical Center Utrecht, Utrecht University, Universiteitsweg 100, 3584 CG Utrecht, The Netherlands; 2Athena Institute, Faculty of Science, VU University Amsterdam, De Boelelaan 1085, 1081 HV Amsterdam, The Netherlands; 3Department of Epidemiology and Data Science, Amsterdam Public Health Research Institute, Amsterdam UMC, VU University Amsterdam, De Boelelaan 1089a, 1081 HV Amsterdam, The Netherlands

**Keywords:** nudging, choice architecture, food choice, socioeconomic position

## Abstract

Nudging has received ample attention in scientific literature as an environmental strategy to promote healthy diets, and may be effective for reaching populations with low socioeconomic position (SEP). Therefore, the objective of this study was to investigate how the determinants of food choice shape the perceptions regarding supermarket-based nudging strategies among adults with low SEP. We conducted semi-structured interviews among fifteen adults with low SEP using a pre-defined topic list and visual examples of nudges. Interviews were recorded and transcribed verbatim and content analysis was used to analyse the data. The results show that food costs, convenience, healthiness, taste, and habits were frequently mentioned as determinants of food choice. However, the relative importance of these determinants seemed to be context-dependent. Interviewees generally had a positive attitude towards nudges, especially when they were aligned with product preferences, information needs, and beliefs about the food environment. Still, some interviewees also expressed distrust towards nudging strategies, suspecting ulterior motives. We conclude that nudging strategies should target foods which align with product preferences and information needs. However, the suspicion of ulterior motives highlights an important concern which should be considered when implementing supermarket-based nudging strategies.

## 1. Background

Obesity presents a global threat to public health: in 2016, more than 1.9 billion adults suffered from overweight and obesity worldwide [1], predisposing them to a range of non-communicable diseases, including type 2 diabetes and cardiovascular disease. The burden of overweight and obesity demonstrates a strong social gradient. In the Netherlands, those with the lowest educational attainment—a key indicator for socioeconomic position (SEP)—are 1.6 times more likely to be overweight and 2.7 times more likely to be obese, compared to people with the highest educational attainment [2] and similar inequalities have been observed in other European countries [3]. Addressing social inequalities is an important step to improving general public health.

One of the major drivers of overweight and obesity is an unhealthy diet, characterized by high intakes of nutrient-poor and energy-dense foods [4], stressing the need for improvement of diet quality, especially among populations with low SEP. However, not all interventions that aim to improve healthy food choices are equally effective across levels of SEP. Individual-based interventions (e.g., mainly focusing on information provision) preferentially improve healthy eating outcomes in individuals with high SEP [5], whereas interventions focused on creating healthier food-environments seem to affect all SEP levels equally [5,6], and could be instrumental in reducing the social inequalities in the burden of overweight and obesity.

Nudging in particular has become an increasingly popular strategy in creating healthier food environments. Nudges make healthy choices easier or more intuitive, without constraining choice for unhealthy alternatives or using financial incentives, for example by placing healthy foods at eye level. Although a wide range of nudging strategies have been applied in various contexts, such as supermarkets and cafeterias [7,8,9,10,11], they often follow a one-size-fits-all approach. The challenge is that customers often constitute a heterogeneous population [12], with potentially differing responses to nudges based on their determinants of food choice (e.g., food values, taste preferences, cooking habits). In order to tackle this issue, recent literature has explored the potential of choice and delivery personalization, which in fact uses these heterogeneity data in order to determine which nudge and which outcome (e.g., food product) will be most effective for targeting a certain group of individuals [13]. Thus, to maximize the potential effectiveness of nudges in populations with low SEP, we need to understand the determinants of food choice in this group, in order to improve the design of nudges so that they fit with their habitual food behaviour practices.

Previous research has reported food costs, familiarity, habits, role models, and food outlet availability as important determinants of food choice among populations with low SEP [14,15,16,17,18], but evidence is relatively scarce. Furthermore, although there is evidence to suggest that people perceive nudging strategies to be acceptable [19], little is known about perceptions regarding supermarket-based nudging strategies among adults with low SEP and how determinants of food choice—such as food costs and habits—may shape these perceptions.

The SUPREME NUDGE project aims to improve lifestyle behaviours and lower cardiometabolic risk among adults with low SEP through implementing nudging and pricing strategies in a real-life supermarket setting [20]. For the design of the nudges in this project, we aimed to explore how the determinants of food choice shape the perceptions regarding supermarket-based nudging strategies among adults with low SEP, in order to better inform the design of the nudges.

## 2. Methods

### 2.1. Participant Recruitment

Participants were recruited in two low-SEP neighborhoods—defined as having a SEP-score below the national average—in the city of Utrecht during two rounds of data collection. During the first round (October and November 2018), participants (*n* = 14) were recruited in a shopping centre in the city district of Overvecht. Of the fourteen participants who were recruited, two participants were not eligible for inclusion in the study based on their educational level and therefore are omitted from data analysis. A preliminary analysis of the data was conducted in order to grasp whether data saturation had been reached. As some new topics emerged from the last interviews that prompted further exploration, additional participants (*n* = 3) were recruited in a community centre in the city district of Rivierenwijk. Formal assessment of thematic saturation using the methodology described by Guest et al. [21] showed that, by using a base size of 4, we reached the 0% new information threshold at 12 + 1 interviews, indicating that during the 13th interview, no additional relevant themes were identified.

During the first round of data collection, the researcher approached shoppers and briefly explained the purpose of the interview. If shoppers were interested in participating in the interview, the researcher immediately invited the participant to a nearby restaurant where the interview took place. Reasons for refusing to participate were mainly related to time concerns (e.g., people were busy doing their groceries). During the second round of data collection, the researcher approached potential participants during an informal coffee morning in a community centre among mostly elderly community members and volunteers. If they expressed interest in participating in the interview, the researcher immediately invited them to a quieter table in the community centre where the interview took place. During the second round of data collection, none of the approached potential interviewees refused to participate. Table 1 provides an overview of participants characteristics.

### 2.2. Interview Procedure

First, the purpose of the interview was once again explained and informed consent was obtained. Prior to the start of the interview, we asked the participants to fill out a short questionnaire on demographic characteristics, including age, sex, and highest attained educational level. Answer possibilities for educational level were based on the Dutch education system, and lower educational level was defined as primary education up to completing intermediate vocational education. This short questionnaire was used as a screener for the researcher, in order to verify that the participants had a lower educational level which was used as a proxy for SEP. As such, the recruitment of low-SEP participants was achieved in two ways: first, by recruiting in areas which had a low neighbourhood SEP-score, and second, by verifying the highest educational level attained by the participant. If the highest attained educational level was not within the definition of the low educational level, the interview was shortened. All participants were reimbursed with a gift voucher worth EUR 20, including the participants (*n* = 2) who did not have a lower educational level.

We conducted semi-structured interviews that lasted on average 25 min, following a pre-defined topic list (Appendix A). The topic list was used in a flexible manner by the researcher, based on the issues that were mentioned by the participants. The first part of the interview was focused on the everyday life of participants, in order to get a feeling of the relevant wider contextual factors that shape dietary choices (e.g., employment, living situation) and in order for the interviewee to feel at ease. In the second part of the interview, the participants were asked how they valued food, and what they perceived to determine their food choices. First, the participants were given the opportunity to answer freely and in their own words, and if deemed appropriate, the researcher probed by suggesting other potential determinants of food choice. In the third part of the interview, we asked under which circumstances it was easy or difficult to eat healthy for the participant. If appropriate, the researcher referred back to what was mentioned in the first and second part of the interview in order to probe further.

In the last part of the interview, the researcher showed various photos of nudges that could be applied in supermarkets to promote healthier food choices, and briefly explained them (Appendix A). Instead of using the term of nudging, we introduced the photos as strategies that supermarkets could implement in order to assist customers in making more healthy food choices. Thereafter, we explored the relative judgement of the nudges by asking participants to choose the photo that appealed to them the most, and we also asked them to explain why. Vice versa, we asked participants to choose the photo which appealed to them the least, and explain why this was the case.

The study was conducted in accordance with the Declaration of Helsinki and was approved by the ethics committee of the University Medical Center Utrecht. Reporting of this qualitative study follows the guidelines set out by the Consolidated criteria for reporting qualitative research (COREQ) checklist [22].

### 2.3. Research Team

The first author (female PhD candidate in the field of epidemiology) conducted twelve of the fifteen interviews. The third author (female PhD candidate in the field of lifestyle epidemiology) conducted three of the fifteen interviews. The interviewers piloted the topic guide together in order for it to be used in a consistent manner. Both interviewers had no prior relationship with either of the interviewees. Interviewees were informed on the scientific background of the interviewers and personal motivations for conducting the interviews (e.g., learning how to help people make more healthy choices in supermarkets) were made explicit. The second author is a male PhD candidate conducting research in the field of systems innovation and transition theory using qualitative research methods, and assisted the first author in performing data analysis and data interpretation. The other members of the research team are (assistant) professors in the field of (lifestyle) epidemiology, and were involved in designing the study, data interpretation, and writing of the manuscript.

### 2.4. Data Analysis

All interviews were audio-recorded and transcribed verbatim by a research assistant and a professional audio transcription company, and all personal identifying information was removed from the transcripts. Transcripts were imported in Atlas.ti (ATLAS.ti Scientific Development GmbH, Berlin, Germany). The process of coding was informed by the Qualitative Analysis Guide of Leuven (QUAGOL) [23], and was based on content analysis using a combination of inductive and deductive coding. First, the transcripts were read and re-read in order for the first author to become familiar with the data. This phase included underlining relevant passages and noting emerging themes in the margin of the transcript. In the second phase, the key elements of the transcripts were summarized in ‘narrative interview reports’, which aimed to holistically capture the experience of the participant. In the third phase, key elements that emerged from the transcripts and narrative interview reports appearing to be relevant for answering the research questions were translated into more abstract concepts. These emerging concepts were used to construct the initial codebook (e.g., inductive coding). For the first research question, this inductive approach was complemented with a deductive approach, by also including relevant concepts identified from literature (e.g., deductive coding) [24,25]. For the second research question, the codebook only included concepts that emerged from the narrative interview reports, thus following a primarily inductive coding approach, as research and theory concerning perceptions, needs, and preferences on nudging interventions are scarce. After coding the first five interviews, the codes and quotations were carefully examined and compared by the first and second author, and minor revisions to the codebook were made. The following five interviews were coded using the updated codebook, and the first five interviews were again analysed in order to adjust the coding following the revisions to the codebook. This iterative process of comparing codes and quotations among the first and second author was repeated until all interviews were analysed. Codes were structured into intrapersonal, interpersonal, socioeconomic and environmental determinants of food choice, following socio-ecological models of health [24]. In order to provide an indication of the relative importance of the identified determinants of food choice, the narrative report of findings is accompanied by a bar graph, indicating the number of interviewees mentioning the specific determinant (interview frequency), and the frequency of coding this determinant in the interview reports (quotation frequency). 

## 3. Results

### 3.1. Determinants of Food Choice

#### 3.1.1. Intrapersonal Determinants of Food Choice

#####  *Physiological Factors* 

We identified three physiological determinants of food choice: ill health, hunger and satiety, and taste preferences (Figure 1). Eight interviewees shared various experiences about ill health, including coping with cancer, diabetes, heart failure, and physical disability. Interviewees said that these conditions influenced their food choices in various ways, for example through affecting taste perceptions, requiring dietary adjustments, affecting product choice, or triggering stress-related unhealthy snacking. Some interviewees also expressed that their own or a relative’s health motivated them to eat healthy:

“Despite being this unhealthy already I still want to reach the age of 100. And then I think, well maybe, despite all these illnesses, [making healthy dietary choices] helps me in achieving that.”(R5)

The role of hunger and satiety was mentioned by five interviewees. Interviewees talked about hunger with respect to meal-timing, for example, a preference for skipping breakfast due to not feeling hungry in the morning. For two other interviewees, this theme was related to food choice, as they perceived some foods as more satiating than others. The role of taste preferences was mentioned by eight interviewees. Taste preferences were often phrased as if they were part of one’s identity, and they often expressed very definite likes and dislikes for various foods:

“I am more of a bread eater” (R3) and “I am not really a fruit type of person”(R2)

#####  *Attitudes, Beliefs, and Perceptions* 

Attitudes towards food varied and could co-occur. Five interviewees had a functional attitude, mainly describing food as a source of energy and nutrients, which are necessary for the body to function. Four interviewees held a hedonic attitude, mainly describing how much they enjoyed food. Five interviewees had a complicated relationship with food by describing that food choice required elaborate contemplation, which was prompted by their health condition or other goals, such as weight loss:

“Since I have heart failure, I have to carefully watch my potassium intake […]. So that is really difficult for me. Because I would love to make a big pan of tomato soup. But I cannot do that anymore”(R5)

Interviewees held various beliefs regarding properties of foods, which influenced their food choices. Four interviewees expressed that food should be of ‘good quality’, referring to the shop where they bought their food (e.g., fish at a local fish shop), the physical appearance of food, or preferences for certain brands. Seven interviewees expressed to prefer fresh foods (e.g., mostly fresh fruits and vegetables), because they perceived this to have better taste, structure, and to be healthier. For some interviewees, these factors outweigh the perceived higher cost or longer preparation time. Finally, five interviewees expressed a general distrust towards the food industry, which was perceived to ‘trick’ consumers or ‘mess’ with the foods.

“Vegetarian food. Who says that is healthy? Maybe they add all kinds of chemical stuff. There is no other way!”(R13)

#####  *Habits* 

Nearly all interviewees framed food choices as habits that they were ‘just used to’ or that ‘worked out’ for them. Interviewees usually reported a traditional Dutch dietary pattern, characterized by potatoes, vegetables, fruits, meats, fish, bread, dairy, and some simple international dishes, such as fried rice and pastas. Some interviewees reported a distinction in food habits between weekdays, weekends, and holidays. During the latter two, interviewees allowed themselves to make more unhealthy food choices. Skipping breakfast was a recurring food habit, mentioned by four interviewees. Despite the ‘static’ notion of habitual food choices, six interviewees had an open attitude towards trying out new foods. Motivations to try new foods included perceived health benefits (e.g., buying spice mixes without salt because of salt restriction), expectations of good taste, or as a means of dietary variation.

#####  *Motivation and Values* 

Food choices were influenced by several commonly valued characteristics: perceived food costs, convenience, healthiness, and taste. Animal friendliness was also mentioned by one interviewee. These values were often weighted against each other when interviewees faced certain food choices. As such, their relative importance was not static, but rather context-specific.

Food costs were mentioned by ten interviewees. Interviewees who valued (low) food costs highly, perceived it to be a waste of money to spend too much money on food, over other ‘better purposes’. One interviewee also expressed to find it important to teach his children that healthy eating could be affordable. Participants who did not value food costs, generally found other food values (e.g., animal friendliness, taste) to be more important.

Convenience was mentioned in ten interviews. Interviewees generally found it important that meals had a shorter preparation time and required a limited number of ingredients. Three interviewees mentioned that convenience was especially important when they had to cook for themselves; when cooking for others, they were prepared to put more effort into cooking. Interviewees generally appreciated convenience in terms of products sold in supermarkets that aided in preparing convenient and healthy meals (e.g., meal boxes or pre-cut vegetables).

The healthiness of foods was mentioned by thirteen interviewees. Only one interviewee expressed to not value the healthiness of foods; healthy foods were regarded as not being tasty. Other interviewees generally stated that they valued the healthiness of foods, which they often substantiated by explaining how they found it important to eat (fresh) vegetables, vitamins, drink water, and to minimize the intake of pork, additives, fat, sugar-sweetened beverage sand sugar. However, the competing value of taste (mentioned by thirteen interviewees) seemed to be a boundary condition for choosing healthy foods, as participants indicated that foods should not be ‘bland’, but should be ‘enjoyable’.

#####  *Knowledge and Skills* 

Interviewees generally seemed familiar with what a healthy diet constitutes, which they described as eating plenty of vegetables, vitamins, fruit, fish, and minimizing intake of sugar, alcohol, and sugar-sweetened beverages in twelve of the interviews. Four interviewees also expressed concerns about additives and artificial sweeteners present in food, particularly in packaged foods and ready-made meals, which they considered to be detrimental to health.

Eight interviewees mentioned various media sources from which they obtained information about nutrition, including television shows, the internet, and doctors. For example, one interviewee described that she had received medication for her cholesterol, which prompted her to search online for dietary regimens:

“And then I read that the combination of cinnamon and honey reduces cholesterol levels. So every morning I add that to my cereal.”(R5)

Three interviewees perceived nutrition information to be confusing, due to conflicting statements and a lack of clarity regarding which sources are credible.

“And I have heard that butter is better than margarine. So yeah, there are just so many perspectives.”(R6)

Five interviewees expressed that they had sought professional help for their dietary behaviours from dieticians or health care institutions, or indicated to have participated in a weight loss program or group therapy for binge-eating. Moreover, five interviewees indicated they were adhering to a diet, such as a low-carb diet or a salt-restricted diet. Furthermore, six participants described various strategies or skills, which facilitated them in making healthy dietary choices. Some strategies were mainly driven by health values, such as making sure that in-home food availability was healthy, restricting snacking after a certain time, or only allowing oneself to fill up their plate once. Other strategies, however, were not only driven by health motives, but also incorporated other values, such as convenience and foods costs. These included: buying foods on offer in large quantities, and/or making meals ahead that one would keep in the freezer, and restricting (unhealthy and unplanned) food purchases by using a grocery list.

#### 3.1.2. Interpersonal Determinants of Food Choice

Partners, family members, neighbours and friends were mentioned as actors influencing food choice in either direct or indirect ways. Partners and direct family members often exerted a direct influence on food choice as they provided input on dinner choices, which was mentioned by nine interviewees. Parents of interviewees exerted more of an indirect influence, as interviewees often perceived certain food habits (e.g., choosing for fresh vegetables) to originate from their upbringing. The composition of the household also influenced food choice, as interviewees were generally more likely to put more effort in preparing meals when they lived or had dinner with others, which was mentioned by three interviewees. Similarly, eating was also often associated with social activities by nine interviewees, such as gatherings with friends and family. Often, these occasions were associated with choosing more unhealthy foods.

#### 3.1.3. Socio-Economic Determinants of Food Choice

Two interviewees explicitly mentioned that they had limited income, which caused them to carefully watch what they spend on food purchases:

“I am price conscious. I have to be, because I only have state pension (AOW)”(R4)

Two other interviewees mentioned that they were brought up with limited financial possibilities, which also affected their perceptions on the affordability of food and food behaviours at present:

“Look, I come from a big family where you could not do just whatever you wanted. And that is also what I want to teach my children. It does not have to cost a lot of money to live healthily.”(R1)

“I have never had a big appetite […] Back in the days, my parents made dinner, and that was split up. So you could never have a second plate. […] That was in a time with limited possibilities”(R9)

Strikingly, as the first quote illustrates, a limited income was not necessarily perceived as a barrier to eating healthy, and the portion restrictions imposed by growing up in a family with limited income actually may have been a facilitator to portion control. Employment also influenced food choices and was mentioned by five interviewees, and primarily exerted influence through the (limited) amount of time interviewees perceived to have available for meal preparation and food shopping. For two other interviewees, (anticipated) employment provided motivation to eat healthy, given a physically demanding job, or because employment was considered to give structure to the interviewee’s daily routine:

“A job would really give some structure to my day. Right now there is just no structure to my day. […] You just don’t take the time to sit down and eat. Then it’s just quick, I will pop by the supermarket and have a croissant.”(R14)

#### 3.1.4. Environmental Determinants of Food Choice

Twelve interviewees mentioned that food products that were on offer in supermarkets often guided their decisions on their purchases. These foods were sometimes bought in large quantities with which interviewees prepared meals in advance. Moreover, interviewees shared various price comparisons for fruits and vegetables available in the supermarket, including the price of organic vs. regular vegetables, fresh vegetables vs. canned vegetables, freshly squeezed orange juice vs. pre-packaged orange juice, or comparing prices amongst food outlets (e.g., discount supermarket vs. regular supermarket). One interviewee also expressed a dislike of the seemingly increasing prices of fresh fruits and vegetables.

“But if I see that snacks are cheap en that fresh food is only getting more expensive… And if you only receive social assistance benefits. That is just undoable”(R14)

Moreover, eight interviewees described the food environment—and particularly the supermarket environment—as tempting, inducing unhealthy food purchases. One interviewee explicitly mentioned that he was aware of some ‘smart’ marketing strategies employed by supermarkets, including secondary placement of food items. In this light, six participants described various strategies which facilitated healthy food purchasing decisions in the supermarket, including making a grocery list prior to going to the supermarket, or shopping for groceries on a weekly basis instead of daily, and thereby minimizing the temptation to unnecessarily buy unhealthy foods.

### 3.2. Perceptions of Nudging

The closing parts of each interview explored interviewees’ perceptions regarding a number of nudging strategies, by means of visual examples (Appendix A). Generally, interviewees had positive perceptions towards the nudges shown during the interviews. More specifically, interviewees seemed to perceive nudges more positively when these were aligned with their perceived determinants of food choice, e.g., with their product preferences, information needs, and beliefs about the food environment.

#### 3.2.1. Product Preferences

Generally, nudges were perceived more positively if they nudged foods which aligned with interviewees’ product preferences in terms of taste, healthiness, costs, and preparation time. Similarly, product characteristics that seemingly made a nudge less appealing were perceived non-freshness, expected poor taste, high costs, and complicated or lengthy preparation. For example, an interviewee indicated to dislike a nudge applied to nuts, because she did not like the taste of nuts. This may suggest that a nudge is less likely to affect a consumer if the nudged product itself does not fit their or their family’s food preferences.

“It’s just as with this: do I find it tasty? Do I like it? Wat does my partner like, that’s also something you need to consider. So, it quickly becomes for me that if there’s something I like, I take it, and otherwise I skip it completely.”(R2)

“But this appeals to me, those recipes. But then it has to be somewhat easy, or saltless.”(R5)

#### 3.2.2. Knowledge and Information

Generally, nudges were perceived more positively if they anticipated on some kind of information need, for example, regarding calorie content and other nutritional information. At times, this need was linked to ill health and dietary restrictions (e.g., a sodium-restricted diet), which prompted a need for individualized dietary information. As an exception, one interviewee indicated that a specific type of information (product popularity) was of very low interest to them. Finally, four other interviewees expressed a certain extent of distrust towards nudges presenting such information, or indicating ‘healthy’ products, suspecting ulterior motives of the supermarket or food industry. As such, these types of nudges will need to be designed in a way which strengthens their credibility, or risk not being trusted by part of the target group.

“The first thing that comes to my mind is: who says this and who gets to decide on this. You get what I mean? At a first glance, this all seems very nice, but then it gets fiddled with by Coca Cola, and they make something different out of it so it does not contain sugar, but then it does contain something else.”(R3)

#### 3.2.3. Food Environment

As previously described, some of the interviewees held rather negative perceptions towards the food environment in supermarkets, which they perceived as tempting and counter-productive in making healthy food choices. In light of this, several interviewees expressed that nudges would actually provide them with the freedom of autonomous choice between healthy and unhealthy foods, illustrating intervention acceptability:

“They should put this on more products. So if you are in front of the crisps shelf, you can think, well, I can take Lays because that is easy. But that then you have an alternative next to it, from which you can see, well, it does actually provide less calories and it is just as tasty.”(R2)

Additionally, it appeared that interviewees had different strategies for navigating the tempting supermarket environment: some adhered to a grocery list or described themselves as habitual shoppers, while others let in-store food availability guide their food purchases. This shopping strategy also influenced interviewee’s perception of nudges: when interviewees described themselves as habitual shoppers they had less positive perceptions towards nudging strategies as compared to interviewees who let in-store food availability guide their food purchases.

## 4. Discussion

The present study aimed to explore how the determinants of food choice shape the perceptions regarding supermarket-based nudging strategies among adults with low SEP. Overall, interviewees mentioned a wide range of factors on an individual level, socio-economic level and environmental level to influence their food choices. Food costs, convenience, healthiness, taste, and habits were frequently mentioned, and their relative importance seemed to be context-specific. Moreover, interviewees generally had positive perceptions towards nudges, especially when they aligned with their product preferences, information needs, and beliefs about the food environment. Still, a small number of interviewees also expressed distrust towards nudging strategies, suspecting ulterior motives of supermarkets or the food industry, irrespective of their determinants of food choice.

### 4.1. Determinants of Food Choice and Nudging

In the present study, we confirm earlier findings on the importance of determinants, such as food costs [14,15,16,17,18,26], convenience [15], healthiness [15,26], and taste [15,26]. Additionally, this study provides preliminary evidence that nudges that align with interviewee’s determinants of food choice, are perceived more positively. This was most notably observed for product preferences: if nudges targeted foods which interviewees liked (e.g., in terms of taste, convenience), then nudges were also perceived more positively. This may imply that the type of food which is nudged may be instrumental for nudge effectiveness in populations with low SEP. For example, following international dietary recommendations, the intake of red and processed meat should be reduced [27]. From the present study, it may be deduced, that in order to nudge people from meat to another substitute, this alternative should align with values such as such as taste and convenience. As a result, nudging a healthy meat substitute (e.g., similar taste and preparation method) may be a better option than, for example, tofu (e.g., notably different taste, potentially different preparation method). We also noted that nudges were perceived more positively when they tapped into certain information needs, which may imply that information nudges [28] may be of particular interest to populations with low SEP.

However, a number of interviewees also voiced concerns about the legitimacy of supermarket-driven health-promoting initiatives, suspecting ulterior motives. Distrust of information on health behaviours has previously been highlighted as a barrier for attaining behaviour change in community-based interventions among populations with low SEP [29] but has not been documented in the specific case of supermarket-based nudging strategies. As opponents of nudging often frame nudging as infringement on autonomous choice [30], one way to address the underlying feeling of distrust is for supermarkets to be transparent about applying nudging strategies using positive framing and to explain why, how, and with whom (e.g., governmental agencies) they participate in such initiatives. In particular, with regard to the application of information nudges, this would imply that the information provided is perceived to come from a credible source. Still, it must be noted that interviewees did not voice ethical concerns about nudging per se, and some even perceived nudges to actually provide them with autonomy to make in-store food choices.

Beliefs about the food environment also influenced perceptions of nudging strategies. We observed that interviewees often viewed the supermarket environment in a relatively negative way, describing it as a tempting environment for which the interviewees had developed strategies to withhold themselves from engaging in alluring unhealthy purchases (e.g., restricting supermarket visits). As such, interviewees perceived nudging to be a welcome strategy for making the supermarket food environment healthier. This contrasts heavily with the perception among retailers of customer demand for healthy foods to be low and customer interest in health to be limited, which is often cited as a barrier for retailer’s engagement in health promoting interventions [31]. This could indicate a mismatch between consumer needs, and retailers’ perceptions of them, and present a potential argument to convince retailers to invest more into health-promoting initiatives.

### 4.2. The Role of Habits

The role of habit as an important determinant of food choice in adults with low SEP, warrants some extra consideration in the context of nudging. Nearly all interviewees indicated that their food habits and/or shopping habits were largely habitual. Habits have been defined as behaviours that have become established through a history of systematic repetition and reinforcement, and therefore have become automatized [32]. Since habits and nudges share the same proposed working mechanism of automated unconscious cognitive processes, the question arises to what extent nudges are able to overrule hard-wired automated habitual food choices. Emerging literature suggests that strong habits or a priori preferences indeed may form boundary conditions for nudge effectiveness regarding food choice [33,34]. For example, a study conducted by Venema et al. showed that the effects of a portion size nudge aimed at decreasing sugar consumption in tea were less effective in participants with strong habits concerning the addition of sugar to tea beverages, compared to participants with less strong habits [34]. Similarly, in a study aiming to nudge participants to a smaller soda drink, it was found that a priori preferences for soda drinks (e.g., liking of those drinks or intentions to reduce consumption of soda drinks), were stronger predictors of the chosen drinks than the nudge itself [33]. Although these former studies were conducted in laboratory settings, these findings may suggest that also in real-life environments such as supermarkets, a priori preferences of customers for specific foods are likely to compete with nudges. Examples of strong preferences that were noted in the present study include the preference for fresh vegetables as opposed to canned vegetables, preference for certain brands, or preference for foods that are convenient. A practical implication that may follow from this, is making sure that the nudged foods are not subject to strong a priori preferences, but represent foods for which adults with low SEP are likely to be fairly indifferent.

### 4.3. Needs Concerning Other Public Health Interventions

The needs expressed by interviewees also underline the importance of other public health interventions other than nudging strategies that aim to promote healthy food choices. In particular, in the case of ill health, there was a need to obtain objective and individualized dietary information (e.g., information on salt content prompted by heart failure induced salt restriction), which may be fulfilled by the use of mobile apps that can be used in the supermarket environment. An example of such an app is the ‘Kies ik Gezond?’ (do I make a healthy choice?) application from the Dutch Nutrition Center, which allows users to scan barcodes in the supermarket and subsequently informs the user whether the scanned food item is contained in the Wheel of Five (the Dutch dietary recommendations), and also provides healthy alternative suggestions for unhealthy foods. The use of these apps could be promoted, especially given the fact that some interviewees perceived nutrition information to be conflicting, and even obtained very erroneous nutrition information from sources, such as the Internet. Lastly, the interviews highlighted that price of foods often guided food choice within the supermarket setting, indicating that pricing strategies are likely to be supported by populations with low SEP. Concluding, the needs expressed by the interviewees suggest that nudging is not likely to be a silver bullet for achieving healthy diets [35], but that in fact, we need a multitude of interventions (e.g., personalized nutrition advice, pricing strategies) to match the varying needs and preferences of adults with low SEP.

### 4.4. Limitations and Strengths

The present study should be interpreted in light of some limitations. First, we only used educational level as an indicator of SEP, as we deemed it stigmatizing to ask interviewees about other proxies of SEP such as income. As it is well-known that SEP is a multidimensional construct, the use of educational level only is likely to be a simplification of reality. Moreover, given the difficulties associated with recruiting individuals with low SEP [36], we adhered to a broader definition of lower educational than, for example, the definition used by the Dutch Central Bureau of Statistics, for pragmatic reasons. Third, our sample was characterized by a relatively high age, which may limit the generalizability of our findings to younger populations.

The study also had several strengths. The analysis and presentation of the data were thoroughly reviewed by and discussed with a second researcher, which strengthens the internal validity of the study. Although not systematically recorded, the interviewers judged the interviewees to constitute a representative sample of the population with regard to weight status, which may be of importance when exploring the topic of food choice. Lastly, although the sample size was relatively small, saturation was reached, which indicates that all major determinants for the interviewed group are likely to have been identified.

## 5. Conclusions

Food costs, convenience, healthiness, taste, and habits were perceived to be important determinants of food choice among adults with low SEP. Interviewees especially appreciated nudges when they targeted foods that aligned with their product preferences, information needs, and beliefs about the food environment. These underlying preferences can be taken into account in the design of nudges, in particular for choosing the type of food to be nudged. Additionally, given the variation in determinants of food choice and their context-specific relevance, applying a wide variation of nudging strategies may also be viable, since they may appeal to a wider audience. Last, some interviewees also voiced concerns about the legitimacy of nudging strategies suspecting ulterior motives, which may undermine nudge effectiveness and should be considered when implementing supermarket-based nudging strategies.

## Figures and Tables

**Figure 1 ijerph-18-06175-f001:**
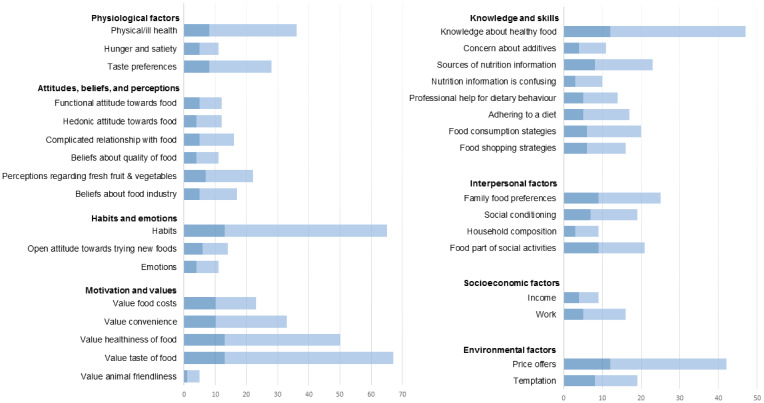
Interview and quotation frequency of perceived determinants of food choice.

**Table 1 ijerph-18-06175-t001:** Participant characteristics.

Interviewee #	Age	Sex	Round of Data Collection	Educational Level
Interviewee 1	33	male	1	intermediate vocational education
Interviewee 2	51	female	1	intermediate vocational education
Interviewee 3	46	female	1	intermediate vocational education
Interviewee 4	68	female	1	advanced elementary education
Interviewee 5	69	female	1	lower vocational education
Interviewee 6	54	male	1	lower vocational education
Interviewee 7	70	male	1	intermediate vocational education
Interviewee 8	71	female	1	lower vocational education
Interviewee 9	79	male	1	advanced elementary education
Interviewee 10	65	female	1	intermediate vocational education
Interviewee 11	58	female	1	intermediate vocational education
Interviewee 12	79	female	1	intermediate vocational education
Interviewee 13	73	female	2	advanced elementary education
Interviewee 14	57	male	2	lower vocational education
Interviewee 15	50	female	2	advanced elementary education

## Data Availability

The data presented in this study are available on reasonable request from the corresponding author.

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
