# Peer review of "Determinants of Food Choice and Perceptions of Supermarket-Based Nudging Interventions among Adults with Low Socioeconomic Position: The SUPREME NUDGE Project"

_ijerph, 2021, doi:10.3390/ijerph18116175_

Round 1

Reviewer 1 Report

Thank you for the opportunity to review you paper: Determinants of Food Choice and Preferences for Supermarket-Based Nudging Interventions Among Adults with Low Socioeconomic Position: The SUPREME NUDGE Project.

The paper describes a sound methodology, and rich qualitative results, however, my biggest concerns is whether the approach you have taken, or whether the things you have found actually meet the purpose you stated in the introduction.  This mismatch arises because nudging is (by your definition) intuitive, and your approach requires participants to be 'deliberative' which is the opposite. Philosophically, and conceptually, asking people to deliberate about how they would like to be intuitive does not make sense.

In terms of this paper - I feel the pathway you should take in revision is reframing your introduction and discussion.  There is value in finding out more about the determinants of food choice in this group for the purposes of designing better nudges. What you have found will help you design more suitable nudges - not because it is how those individuals prefer to be nudged - but because you are creating easy pathways, shortcuts, defaults that are aligned with the way they operate/think/act usually. 

In the introduction, this means rethinking the last paragraph (page 2 lines 72-89). In particular, removing the word co-creation (co-creation is generally understood to mean the design of programs/services/strategies together with users based on the ideas and initiatives of the participants - not just talking about past nudges to gather their opinions of them). I see no evidence of co-creation or co-design in this paper - so best to remove this word so that you do not confuse the reader. Then, reframe the last few lines to state that you are exploring how food determinants may shape the perceptions of nudges, and how together this may lead to the design of better nudges for this group.

In the results, section 3.5, I feel lines 428-429 are key to what you seem to be seeing here (and this matches better with the philosophy and concepts of nudging.  The determinants affected their perceptions of nudges. Because of the way the individuals operate/think/act usually, some nudges appear easier, more natural, more intuitive, to them. Some careful thinking around your findings in this section - through a lens that supports dual-process thinking (system 1/system 2 thinking that underpins most approaches to nudging) - would enable you to reframe/rewrite this section.   

I strongly encourage you to then revisit the discussion section to draw together the links between 1) the determinants of food choice in this group, 2) how food determinants shaped the perceptions of nudges and 3) how this together informs the design of better nudges for this group. This also allows you to reflect on which typical nudging strategies (see systematic literature reviews on nudging and food choice for some concrete examples) would be likely to be most effective in this group - which is currently missing from this paper.

Lastly - do a careful review of the language. There were a few places where is was not smooth - for example, Section 3.4 line 336 "expressed to have sought" should be "expressed that they had sought"; line 339 "indicated to adhere to a diet" should be "indicated they were adhering to a diet". These are just two examples, there are some more throughout. 

I wish you well with your revision and future research. 

Reviewer 2 Report

Harbers et al. reported an interesting study about the determinants of the food choice among the low economic and educational level persons. The aim was the possibility to construct strong, effective and successful strategies nudging the low-economic-level people to consume healthy food. I appreciated, other than the methodology, the location of interviews (e.g., supermarket) to contextualize them and to help the interviewees in the appropriate responses. I agree with the authors conclusions regarding the difficulty to nudge the choices and the need of enlarging the strategies to healthy food purchase. I suggest accepting the manuscript after minor revisions:

  • the word elimentary in Tab.1: is it a mistake or has it a meaning?
  • all the citations are to be corrected and written as requested from the Journal: add the point after name initial letter; do not use et al.; the end page of the articles; no issue number; year in bold type.

Reviewer 3 Report

Comments in the attachment.

Round 2

Reviewer 1 Report

The revision is good, and the manuscript reads more clearly. Apart from the comments below - I feel the framing is now better.

Table 1: elementary is spelled incorrectly as: eleimentary

In many places (possibly entirely throughout), new additions have been added without removing old text. For example - in the next sentence I have struck out a piece that still appears in the revised manuscript (eg Five interviewees expressed to have soughtthat they had sought professional help for their dietary behaviours). This occurs in the title also ('perceptions' should have replaced 'preferences').  Please check insertions throughout.

Reviewer 3 Report

I accept the response to the review.
